# Juvenile Idiopathic Arthritis: Roma Children Seem to Run More Risk than Non-Roma

**DOI:** 10.3390/ijerph17072377

**Published:** 2020-03-31

**Authors:** Simona Drobnakova, Andrea Madarasova Geckova, Veronika Vargova, Ingrid Schusterova, Jaroslav Rosenberger, Daniel Klein, Jitse P van Dijk, Sijmen A. Reijneveld

**Affiliations:** 1Department of Paediatric and Adolescent Medicine, Faculty of Medicine, P.J. Safarik University, Tr. SNP 1, 040 01 Kosice, Slovak Republic; veronika.vargova@upjs.sk (V.V.); ingrid.schusterova@upjs.sk (I.S.); 2Graduate School Kosice Institute for Society and Health, Faculty of Medicine, P.J. Safarik University, Tr. SNP 1, 040 01 Kosice, Slovak Republic; andrea.geckova@upjs.sk (A.M.G.); rosenberger.jaroslav@gmail.com (J.R.); j.p.van.dijk@umcg.nl (J.P.v.D.); 3Department of Health Psychology, Faculty of Medicine, P.J. Safarik University, Tr. SNP 1, 040 01 Kosice, Slovak Republic; 4Olomouc University Social Health Institute, Palacky University, Univerzitni 22, 771 11 Olomouc, Czech Republic; 52nd Intenal Clinic, Faculty of Medicine, University PJ Safarik, 040 01 Kosice, Slovak Republic; 6Institute of Mathematics, Faculty of Natural Sciences, P.J. Safarik University, Tr. SNP 1, 040 01 Kosice, Slovak Republic; daniel.klein@upjs.sk; 7Department of Community and Occupational Medicine, University of Groningen, University Medical Center Groningen, Ant. Deusinglaan 1, 9713 AV Groningen, The Netherlands; s.a.reijneveld@umcg.nl

**Keywords:** juvenile idiopathic arthritis, Roma, ethnicity, Slovakia

## Abstract

*Background:* Ethnic information regarding juvenile idiopathic arthritis (JIA) exists for various populations across the world but is fully lacking for Roma. We assessed the occurrence and clinical characteristics of JIA in Roma vs. non-Roma children. *Methods*: We obtained data on all outpatients (*n* = 142) from a paediatric rheumatology centre (age 3 to 18 years) in the eastern part of Slovakia (Kosice region). We assessed patients’ age, gender, disease type and related extra-articular conditions by ethnicity. We obtained population data from the 2011 census. *Results*: The share of Roma children was higher in the clinical JIA sample than in the overall population (24.6%, *n* = 35, Roma in the sample vs. 10.8%, *n* = 142, Roma in the population, *p* < 0.05). Moreover, Roma children had been diagnosed more frequently with extra-articular conditions but did not differ in other symptoms. Treatments also did not differ by ethnicity. *Conclusion*: Roma children had been diagnosed more with JIA than their non-Roma peers. This calls for further research on the causes of this increased disease burden in Roma children.

## 1. Introduction

Juvenile Idiopathic Arthritis (JIA) is an autoimmune, noninfectious, inflammatory joint disease of more than 6 weeks duration in children under 16 years of age [1,2]. The knee is the most commonly affected joint in JIA [3]. In a systematic review of JIA in 2014, Thiery et al. found that the incidence of JIA varied from 1.6 to 23.0 and the prevalence from 3.8 to 400.0/100,000 across Europe. A higher incidence and prevalence were observed in girls than in boys [4]. Saurenmann et al. found that children with European ancestry have a higher risk for developing JIA in the general population, whereas children of black, Asian or Indian origin have a significantly lower risk [5]. 

Roma constitute a major minority group in several countries in Central and Eastern Europe (CEE), and this holds for Slovakia, too. Genetic studies confirmed Roma originated in the Punjab region of north-western India as a nomadic people. After leaving India, the Roma passed to Europe between the 11th and 12th centuries. For centuries, Roma were scorned and persecuted across Europe [6]. Over half of Slovak Roma (current estimates of their total number is up to 450,000 persons) resides in segregated settlements [7,8], similar to the situation elsewhere in CEE. These settlements are characterized by general poverty, high rates of unemployment and low education, substandard material conditions, access barriers to health care and other public services and a generally poor health status [9,10,11]. The generally poor living conditions of Roma can be expected to lead to worse clinical consequences of JIA and to barriers that cause their entry into health care in more acute states.

Information on JIA in Roma compared to non-Roma children is thus far fully lacking, though based on the above-mentioned literature a significant difference by ethnicity is likely. Therefore, this study aims to assess the occurrence of JIA and the prevalence of extra-articular conditions and clinical characteristics in Roma vs. non-Roma children.

## 2. Materials and Methods

### 2.1. Sample

We recruited patients aged 3–18 years from the outpatient clinic at the Eastern Slovak Centre of Paediatric Rheumatology in the Children’s University Hospital, Kosice. This Centre covers the entire JIA-children population in the Kosice region. Patients were diagnosed by a paediatric rheumatologist from the paediatric rheumatology centre in the Children’s University Hospital, Kosice, and classified based on the International League Against Rheumatism (ILAR) criteria. All patients or when relevant their parents agreed with participation.

Data were collected from 1 October 2017 to 30 June 2018. The study was approved by the Ethics committee of the Children’s University Hospital, Kosice, GCP 135/95, on 20 September 2017.

### 2.2. Procedures and Measures

We obtained data on ethnicity, clinical characteristics (JIA type, extra-articular conditions, type of treatment and disease duration) and background (age, gender) from the medical records. Ethnicity was assessed by self-identification and was compared to the determination judged by a physician, and in case of a mismatch, the opinion of the head-nurse was leading. The existence of an extra-articular condition was, apart from clinical appearance, indicated by the anti-nuclear antibody values (ANA).

Data on JIA types were coded following the ILAR classification system [12]. This includes seven types of JIA: Oligoarticular JIA, seropositive polyarticular JIA, seronegative polyarticular JIA, systemic-onset JIA (sJIA), enthesitis-related arthritis (ERA), juvenile psoriatic arthritis (JPsA) and undifferentiated JIA assessed by a paediatric rheumatologist.

We used ANA testing to assess the extra-articular conditions. This testing was done by indirect microscopic serology (indirect immunofluorescence) using HEp-2 cells as a substrate. To evaluate the results we used the standard of fluorescence patterns intensity of nucleolar staining assessed by one immunologist.

We further obtained data for the share of Roma by age category in the total population from the Slovak Population and Housing Census 2011 of the Statistical Office of the Slovak Republic.

### 2.3. Statistical Analyses

First, we assessed background and clinical characteristics of the sample using descriptive statistics. Second, we compared the ethnic composition of the clinical sample (patients diagnosed with JIA) with that of the full population aged 5 to 19 years and assessed differences using the chi-squared and Fisher’s exact tests for proportions. Finally, we explored the ethnic differences in the prevalence of extra-articular conditions using logistic regression, crude and adjusted for gender and age, and additionally for disease duration. Statistical analyses were performed using SPSS version 21.0 (IBM, Armonk, NY, USA).

## 3. Results

The sample consisted of 142 consecutive JIA patients aged 3 to 18 years (mean ± SD: 11.36 ± 4.72 years; 52 males).

### 3.1. Ethnic Differences in Selected Clinical Indicators of JIA and Its Treatment

The background and clinical characteristics of the sample by ethnicity are described in Table 1. Mean ages hardly differed, whereas disease duration was somewhat shorter in Roma. We further found a roughly equal distribution of the JIA type in Roma vs non-Roma children patients, with relatively most children having the oligoarticular and polyarticular types of JIA. Extra-articular conditions occurred more frequently among Roma than among non-Roma children patients. There were 17.1% Roma (*n* = 6) and 29.9% non-Roma (*n* = 32) patients on biological treatment.

### 3.2. Proportion of Roma Children in the Clinical Sample vs. the Population by Census

The proportion of Roma children was significantly higher in the clinical sample than in the population from the census (Table 2).

### 3.3. Proportion of Roma and Non-Roma Children Having Extra-Articular Conditions

In our sample of 142 JIA patients, 77 patients had Anti-nuclear antibodies. Roma patients had higher odds of diagnosing with extra-articular conditions than their non-Roma peers and girls, respectively (see Table 3).

## 4. Discussion

We assessed the occurrence of JIA and the prevalence of extra-articular conditions and clinical characteristics in Roma vs. non-Roma children, as studies on Roma ethnic differences in this field are fully lacking. We found that Roma children have been diagnosed more frequently from JIA and relatively more frequently had extra-articular conditions. They did not differ in other clinical symptoms and received similar treatment.

We found that Roma children had been diagnosed more frequently with JIA, which is a novel finding, as no previous evidence is available on this issue. This finding could be explained in several ways. First, the incidence of JIA may be higher due to a higher prevalence of various risk factors among Roma in either the environment or in their biological background. During the last ten years, Roma health and also of Roma children has been more intensively studied, and their health outcomes are still very different from those of non-Roma. The psycho-social aspects of discrimination [13], financial deprivation and their statements about their poor health [14] could also play a role regarding higher morbidity. Biological, e.g., genetic, differences are a much less likely explanation [15]. A second explanation may be that prevention, detection, and treatment of the early stages is worse among Roma children. This higher prevalence of JIA among Roma children clearly calls for further research. Despite the fact that health care in Slovakia is free by law, many Roma face barriers to using it or at least reaching it. In addition, Roma have some unique cultural differences and belief related to health and health care [16]. JIA thus occurs more frequently among Roma children, which calls for further exploration.

We found that compared to non-Roma children, Roma children had been diagnosed more frequently with extra-articular conditions of JIA. Differences in health status, health outcomes and prevalence of diseases among different racial and ethnic groups, in general, are well documented. There is a growing number of studies mapping ethnic differences in the incidence and the course of various diseases; however, few of them are related to children [17,18], and those that do exist mostly regard infant mortality [19] or morbidity. Dostal et al. (2010) found that the incidence of influenza, otitis media, intestinal infectious diseases and viral diseases was significantly higher in Roma than in non-Roma children at the age of 0-2 years [20]. Similar to our Roma children studies, Kolvek et al. in 2014 [17] also found a high representation of Roma paediatric patients among primary renal-disease (PRD) patients presenting with obvious signs in comparison to non-Roma. Other studies have pointed to the poor nutritional status in Roma mothers through pregnancy and the dietary and smoking habits of pregnant Roma women leading to lower birth weight in their new-borns. Compared to non-Roma mothers, Roma mothers had different breastfeeding habits and lower blood levels of folate, b-carotene, retinol, and a-tocopherol, leading to shorter pregnancy duration and more spontaneous abortions, abruptions, placental infarctions and congenital defects [21]. In 2008 Chen et al. found an association between high levels of family stress and air pollution on asthma outcomes, including greater production of asthma-related inflammatory markers. Consequently, this combination in Roma of high exposure to prenatal risks and to postnatal risks, such as air pollutants and stressors, may lead much higher morbidity in Roma children [22].

Despite the fact that the exact aetiology of an autoimmune disease such as JIA is unknown, it is believed that a combination of genetic predisposition and environmental factors play role. Progress has been made in identifying the genes underlying JIA susceptibility, but only a few studies have shown the identification of environmental factors to be a trigger of the immunological abnormalities of JIA. Most of these studies discussed the connection between early life infection, maternal smoking, and autoimmune disease, but ethnicity was involved in only very few of them [23,24]. In 2018 Manzano-Gamero et al. found a higher risk of antiphospholipid syndrome in Roma patients diagnosed with systemic lupus erythematodes [25]. Their higher morbidity and social factors, such as unemployment, dependence on social support, poverty and low understanding of health mechanisms, contribute to the development of communicable diseases [26]. All of these triggers may result in the occurrence of a higher prevalence of JIA among Roma, as we found. Ethnic comparisons of childhood rheumatic disease are rarely described [5]. Thus, further research is needed on the differences appearing between morbidity among Roma and non-Roma paediatric patients. 

### 4.1. Strengths and Limitations

Our study is the first to focus on JIA and Roma ethnicity. Its strengths lie in that it comprises a full region as a catchment area and a full clinical assessment of JIA. Some limitations should also be mentioned.

First, our prevalence estimates depend on the share of Roma in the population, which may be too low in comparison to the census from 2011. The proportion of Roma in the population is generally underestimated by the census. Second, our cross-sectional design does not allow us to uncover any causal pathways. 

Our clinical sample—Roma children diagnosed with JIA from the Kosice region—suggests that more Roma children have JIA, which could be expected based on their share in the population in the Census of 2011. This might be due to an underestimation of the number of Roma in the Census. However, this underestimation seems to be too small to explain this finding.

### 4.2. Implications

We found the prevalence of JIA to be much higher among Roma, which implies that more attention should be paid to JIA among this ethnic group. As Roma children are a small group in the population in absolute terms, more analyses need to be done over all of Slovakia or in more CEE countries to confirm the theory of Rambouskova et al. [21] on maternal nutrition. Furthermore, whether the incidence and the course of the illness are associated with differences in Roma vs. non-Roma regarding their living conditions, such as their socioeconomic conditions, parental health literacy, access to health care, the quality of health care provided and adherence to preventive advice, should be studied.

## 5. Conclusions

Our findings indicate that Roma may be diagnosed more with JIA than non-Roma. This is a call for further action also given the growth of the Roma population [21].

## Figures and Tables

**Table 1 ijerph-17-02377-t001:** Ethnic differences in selected clinical indicators of JIA and its treatment.

Variable	Category	Roma	Non-Roma
		*N* (%) 35 (100.0)	*N* (%) 107 (100.0)
Type of JIA	Oligoarticular	15 (42.9)	44 (41.1)
	Polyarticular	13 (37.1)	47 (43.9)
	Enthesitis	4 (11.4)	7 (6.5)
	Systemic	1 (2.9)	4 (3.7)
	Systemic/MAS	1 (2.9)	4 (3.7)
	Psoriatric	1 (2.9)	1 (0.9)
	Undifferentiated	-	-
Extra-articular conditions	Positive	23 (65.7)	54 (50.5)
Uveitis	Positive	3 (8.6)	19 (17.8)
Treatment	sDMARDs	12 (34.3)	32 (29.9)
	bDMARDs	6 (17.1)	32 (29.9)
	Remission	15 (42.9)	40 (37.4)
	NSAIDs	2 (5.7)	3 (2.8)
Duration(years) ^1^		3.77 ± 3.25	4.34 ± 3.75
Age (years) ^1^		11.91 ± 4.70	11.18 ± 4.74

^1^ mean ± SD; MAS—Macrophage-activation syndrome; sDMARDs—synthetic disease-modifying antirheumatic drugs; bDMARDS—biological disease-modifying antirheumatic drugs; NSAIDs—Nonsteroidal anti-inflammatory drug.

**Table 2 ijerph-17-02377-t002:** Share of Roma and all children in the population of the Kosice region and in the juvenile idiopathic arthritis (JIA) patients from the Kosice region, based on the 2011 census and JIA patients for the Kosice region.

Variable	Roma	Kosice Region
	*N*	*N*
All children aged 5–19 years	13,712	126,851
Children diagnosed with JIA aged 3–18 years	35	142
Estimated prevalence JIA/100,000	255.3	111.9

**Table 3 ijerph-17-02377-t003:** Occurrence of extra-articular conditions by ethnicity in children with JIA: proportions and odds ratios (95% confidence intervals, CI) resulting from logistic regression models.

Variable	Categories	*N* (in %)	Crude Odds Ratios (95% CI)
Ethnicity	Roma	23 (65.7)	2.33 (1.00–5.43) *
	Non-Roma	54 (50.5)	
Age			0.94 (0.86–1.02)
Gender	Male	21 (40.4)	2.34 (1.13–4.86) *
	Female	56 (62.2)	
Duration			1.06 (0.96–1.18)

* *p* < 0.05.

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
