# Peer review of "Juvenile Idiopathic Arthritis: Roma Children Seem to Run More Risk than Non-Roma"

_ijerph, 2020, doi:10.3390/ijerph17072377_

Round 1

Reviewer 1 Report

The paper presented by Drobnakov et al. concerns an interesting issue of ethnic differences in the incidence of JIA. The described study certainly has some value, but it is a simple cross sectional study. In my opinion, such a study is too preliminary to be published in IJERPH. In addition, the conclusions and discussion presented are very general. There is no specification of the time when the study was conducted, the authors indicate the year 2017, however the study was approved on 20 September 2017. The authors state the sample consisted of 142 consecutive JIA - according to the data given in Table 2. these are all children from JIA from the Kosice region. The occurrence of JIA in the group of Roma children in the presented study is very high. A study designed in such a way should obtain much more data on factors affecting children's health. In its current form it is only a summary of statistical data.

Author Response

Please see the attachment. I´ve made changes.

Data were collected from the 1st of October 2017 till the 30th of June 2018.

We were facing a deficient database of the proportion of Roma in the general
population. Our sample - Roma children suffering from JIA from the Kosice region - suggests that we have more Roma patients in comparison to the Census from 2011. However, the Census generally underestimates the proportion of Roma in the population. The number of children estimated by Sprocha (2014) and the Atlas of Romani community the Roma population in 2013 over the whole country is around 7.4%, in the eastern part of Slovakia it is around 29.8%. One of the problems is that any estimation of the Roma population in Slovakia does not exactly reflect the current number because of the huge number of the Roma do not register for the Roma nationality officially. This fact probably explains this problem.

Reviewer 2 Report

1) Lines 45-51- There is insightful information provided on the health concerns of the Roma.  It might be helpful to mention some of the antecedents of these health concerns.  There should be a description of the origin, identity, and history of the Roma people (e.g., origin in Punjab region of northern India, being view as nomadic people, history of persecution and discrimination) and how this contributes to health disparities in this group.

2) Lines 68-70- The sentence 'Ethnicity...decisive' is unclear and has awkward wording.  Consider rewording.

3) The Literature Review and Discussion sections are insightful.  In terms of environment risk factors, it would be insightful to comment on any studies that have explored the risk factors (either protective or harmful) of: breastfeeding, secondhand smoke, sun exposure and vitamin D, exposure to pollutants, animal exposures, and stressors.  Further exploration into these factors may complement the socioeconomic factors you discuss in lines 153-155 and also add depth to the discussion.

4) This paper is insightful, compelling, and novel.

Reviewer 3 Report

General Comments

Please review the entire manuscript for consistent use of abbreviations, e.g. juvenile arthritis appears in 2.3 Statistical analysis when JIA should be used.

Abstract

The abstract is clear and provides relevant information. Where percentages are reported in the abstract, these should be accompanied by an N= to give context to the percentage. In terms of language please revise the use of the word “suffer” in the abstract – this links to patient first language and use of the word “suffer” implies that all those diagnosed with JIA do suffer. This also applies to the entire manuscript.

Introduction

The Introduction is very succinct, and although it provides some important information the authors should elaborate to strengthen the reasoning for conducting this research.

Materials and Methods

The determination of ethnicity seems problematic, if the opinion of the rheumatologist and head nurse made a decision for someone who had not self-identified. The authors should explain this further.

The use of the Census 2011 data should also be considered. What was the response rate to the census? Could a lower response rate from the Roma community explain the difference in proportion of children who had JIA in the clinical sample?

Results

Ensure that where a percentage is reported in the Results that a number (N=) is reported to give the reader an understanding of the statistic.

Discussion and Conclusion

The Discussion and conclusion seem fair, based on the information presented earlier in the study. However given the brief nature of the manuscript the Discussion doesn’t allow for an in-depth exploration of the issue.

Tables/Figures

No revisions required. Tables are clearly presented.

Author Response

Point 1: 1) Please review the entire manuscript for consistent use of abbreviations, e.g. juvenile arthritis appears in 2.3 Statistical analysis when JIA should be used.

Response 1: I´ve made these changes. Please see the attachment.

Point 2: Abstract: The abstract is clear and provides relevant information. Where percentages are reported in the abstract, these should be accompanied by an N= to give context to the percentage. In terms of language please revise the use of the word “suffer” in the abstract – this links to patient first language and use of the word “suffer” implies that all those diagnosed with JIA do suffer. This also applies to the entire manuscript.

Response 2: I´ve made these changes. Please see the attachment. I´ll ask once more for language proof.

Point 3: The Introduction is very succinct, and although it provides some important information the authors should elaborate to strengthen the reasoning for conducting this research.

Response 3:

Roma constitute a major minority group in several countries in Central and Eastern Europe (CEE), and this holds for Slovakia, too. Genetic studies confirmed Roma originated in the Punjab region of north-western India as a nomadic people. Genetic studies confirmed Roma originated in the Punjab region of north-western India as a nomadic people. After leaving India, the Roma passed to Europe between the eighth and tenth centuries C.E. For centuries, Roma were scorned and persecuted across Europe . Over half of Slovak Roma (current estimates of their total number is up to 450,000 persons) resides in segregated settlements [6,7], similar to the situation elsewhere in CEE. These settlements are characterized by general poverty, high rates of unemployment and low education, substandard material conditions, access barriers to health care and other public services and a generally poor health status [8,9,10]. The generally poor living conditions of Roma can be expected to lead to worse clinical consequences of JIA and to barriers that cause their entry into health care in more acute states.

Information on JIA in Roma compared to non-Roma children is thus far fully lacking, though based on the above-mentioned literature a significant difference by ethnicity is likely. Therefore, this study aims to assess the occurrence of JIA and the prevalence of extra-articular conditions and clinical characteristics in Roma vs. non-Roma children.

Point 4:

a.) The determination of ethnicity seems problematic, if the opinion of the rheumatologist and head nurse made a decision for someone who had not self-identified. The authors should explain this further.

b.) The use of the Census 2011 data should also be considered. What was the response rate to the census? Could a lower response rate from the Roma community explain the difference in proportion of children who had JIA in the clinical sample?

Response 4:  

a.) Ethnicity was assessed by self-identification and was compared to the determination judged by a physician, and in case of a mismatch the opinion of the head-nurse was conducive.

b.) We were facing a deficient database of the proportion of Roma in the general population. Our sample - Roma children suffering from JIA from the Kosice region - suggests that we have more Roma patients in comparison to the Census from 2011. However, the Census generally underestimates the proportion of Roma in the population. The number of children estimated by Sprocha (2014) and the Atlas of Romani community the Roma population in 2013 over the whole country is around 7.4%, in the eastern part of Slovakia it is around 29.8%. One of the problems is that any estimation of the Roma population in Slovakia does not exactly reflect the current number because of the huge number of the Roma do not register for the Roma nationality officially. This fact probably explains the small significance of our sample.

Point 5:  Ensure that where a percentage is reported in the Results that a number (N=) is reported to give the reader an understanding of the statistic.                                          

Response 5: There were 6 Roma (17.1%) and 32 non-Roma (29.9%) patients on biological treatment.

Point 6:  The Discussion and conclusion seem fair, based on the information presented earlier in the study. However given the brief nature of the manuscript the Discussion doesn’t allow for an in-depth exploration of the issue.

Response 6: Please see the attachment I´ve added a few lines to the text.

Round 2

Reviewer 1 Report

The paper, after the correction made by the authors, can be published.

Reviewer 3 Report

Suggested revisions have been made as required.